# Incorporating Form-Based Codes into the Design-Based Approach to Historic Building Conservation in Phuket, Thailand

**Kanokwan Masuwan and Pusit Lertwattanaruk ***

Faculty of Architectural and Planning, Thammasat University, Klongneung, Khlong Luang,
Pathum Thani 12121, Thailand; kanok.masuwan@gmail.com
**\*** Correspondence: lertwatt@tu.ac.th; Tel.: +66-29869434

**Abstract:** In Thailand, the concept and measurement of urban planning rely on conventional zoning, which includes land use, building usage, and open space ratio. Conventional zoning focuses on both the growth of buildings in terms of physical dimensions and their usability in lowland areas. The guidelines and measures used in urban planning do not reflect the spatial relationship of the community, as they have a less detailed design and place no emphasis on the identity of the district. Urban planning might not protect the sense of any given place, as it often uses a one-size-fits-all plan that is then applied to whole communities. Form-Based Codes (FBCs) are urban planning tools that are used to maximize land use, benefit the public, focus on creating a specific physical form, and design the development and public spaces in a way that matches the community's vision. FBCs are the result of the cooperation between stakeholders, architects, urban planners, government agencies, and members of the local community who are willing to create a plan for their public space and to preserve the physical characteristics of the city. In this paper, we aimed to understand the relationship between various historical contexts and the FBCs using the case study of Phuket's old town, which has a fusion of Sino–Portuguese architecture. Building form standards suitable for Phuket's old town were created by comparing them to a baseline case that uses existing codes and regulations and using the FBCs' components. FBCs have the potential to enhance the character and vibrancy of the historic area by improving façade design and preserving the sense of place and community pride. The results support the hypothesis that FBCs are able to supplement conventional zoning in historic districts. Recommendations for a local historic preservation commission and communities that are considering the adoption of FBCs for historic resources and districts are provided.

**Keywords:** form-based codes; historic conservation; Phuket old town; conventional zoning

## 1. Introduction

Form-Based Codes (FBCs) are urban planning tools that are used to maximize land use, benefit the public, and achieve a specific urban form. The purpose of these tools is to preserve the identity of the area and serve as a measure to contain urban sprawl, through the effective design of public spaces, and to define physical features, shapes, and components of the city [1]. FBCs are a result of the cooperation of all stakeholders, architects, city planners, government agencies, and people willing to create a plan for the public space and the physical characteristics of the city. FBCs integrate development factors and enforce laws in conjunction with other relevant urban plans [2].

### 1.1. Relevance of FBCs

FBCs have been widely applied as urban design tools, with a focus on the effective planning of neighborhoods, towns, and cities not only in the USA but also in Brazil, Australia, China, Ecuador,

China, and Hong Kong. Because FBCs are not a one-size-fits-all solution, we encourage place-making that is appropriate to the city. There has been no implementation of FBCs in ASEAN countries due to the complexity of the existing laws, including property rights and centralized government systems, which promote social inequality. Currently, urban planning systems fall under government control [3–5]. Developing Thailand's urban plan, becoming a pioneer in advanced urban planning policy for problem-solving, and determining a future direction for the ASEAN countries are additional targets.

FBCs define the frameworks and methods required to achieve urban planning goals and community visions, such as preserving the identity of the district. These city development tools are beneficial for the design and preservation of a physical area's characteristics and form. However, FBCs have also benefited designers, urban planners, the government sector, and the public sector, enabling them to deeply understand both a city's visual impact and the relationship among communities. FBCs also play a supporting role in conventional urban planning. Conventional urban planning only considers the benefits of land in terms of general physical form. As a result, it is difficult to identify and predict the growth of the urban form. [6–8].

### 1.2. Components and Process of FBCs

Building standards are the components of FBCs that people use to visualize the identity of the area and are primarily categorized and organized by zone. In this research, we focus on the historic preservation area, which is a special district of the transect zone. The purpose of FBCs is to preserve the character of a district, where building standards are systematically paired with an illustrative plan. Then, a critical analysis of documents and existing regulation can be performed [9,10].

The building standards are as follows:

#### 1.2.1. Building Placement

To regulate the relationship between buildings and the public standards by considering the lot width, the building and ceiling height, the building depth, and the building lot size to understand equal building height, parallel grid lines, repetitive elements, and build-to line (BTL) regulations, whereas conventional zoning controls only the setback, floor area ratio (FAR), building coverage ratio (BCR), and open space ratio (OSR) [11,12].

#### 1.2.2. Allowed Frontage Types

To ensure that the physical form of the frontage will ultimately be consistent with the urban patterns the community wants to replicate and is based on data collected during the documenting process, whereas conventional zoning does not mention frontage types.

#### 1.2.3. Allowed Building Types

The allowed building types are based on a combination of building form and use, determined by the community. Communities are often concerned about potential land uses whose operating characteristics are inherently compatible with the goals and values of the community, being located near highly sensitive areas where there are no land use controls. This also includes pursuing economic development goals through the distribution of specific land uses in various parts of the community.

### 1.3. Problems in Thailand's Urban Planning System

The concept and measurement of urban planning in Thailand rely on conventional zoning, which includes the use of land, open space, transportation, and public utilities. It focuses on the growth of buildings in terms of physical dimensions and their usability in lowland areas. The developers only consider their areas, resulting in uncoordinated development. [13–15].

Top-down development strategies dominate Thai urban planning. The provincial governors, appointed by the central government in Bangkok, the capital of Thailand, were tasked with the

legislation of land use regulations all over the country, which comprises 77 provinces and has a total area of more than 513,120 km$^2$ and a population of 63 million people. The current strategy is inadequate because the central governing bureau lacks personnel, time, and adequate knowledge. Therefore, it is difficult to extract the true potential of a precise area, and there is no communication between the central government and local bureaus, resulting in local bureaus possessing insufficient knowledge about urban planning processes [13,14]. Misinterpretation leads to the violation of these central government-enacted regulations. This shows the importance of public participation in urban planning and legislation. Inattentiveness from the government can contribute to these kinds of problems as well. The community should play a critical role in creating a local identity and designing district development formats that comply with both tangible and intangible district identity values [16–18].

Foreign case study reviews show that essential factors in successful historic preservation all originated from the local community, working in tandem with private sectors and local governments. Before reaching out to the central government, it is important to acknowledge district identity and grant local governments the ability to manage their own historical area. Laws should be in line with this "bottom-up management" style, which is suited to Thailand's current urban planning format [19–21].

### 1.4. The Relationship between the Historic District and the Zoning District

The historic district is a concentrated area of adjacent buildings, which represent a specific period of development, are related by their history, and are architecturally significant [9,22]. The zoning district is a land use planning tool used by local governments to manage the development of land as shown in Table 1.

**Table 1.** The relationship between the historic district and the zoning district [9].

| Topic | Local Historic District (LHD) | Zoning District |
|---|---|---|
| **Focus Point** | Local historic district is based on design review. | Zoning is based on land use. |
| **What Reviewed** | The Historic District Commission (HDC) reviews the proposed work. HDC members gain expertise in architectural styles, historical materials, and the use of appropriate modern materials. | It is reviewed by planning commission members who do not have specialized expertise in historic preservation practices. Places added burden on planning/zoning board members. |
| **Focus** | The design review focuses on the retention of significant details and features that make up the character of a building. | The review focuses on broad-brush issues such as height, bulk, area, density, use, and setback. |
| **Who Reviewed** | The review is based on the Secretary of the Interior's Standards for Rehabilitation, a national set of standards in use for 45 years. | The planning commission/zoning board must develop standards and guidelines. |
| **Related to Preservation Tax Incentive** | The HDC review is based on the same standards developed for federal historic preservation tax credits, which can be used by commercial property owners to rehabilitate historic resources. | The interpretation of standards and guidelines by those unfamiliar with basic preservation principles may result in work that does not meet federal historic preservation tax credit requirements. |
| **What is Public Act and What is Purpose** | The Local Historic District Enabling Act gives local governments the legal authority to regulate work in designated historic districts to increase property values, foster civic beauty, strengthen the local economy, and promote and safeguard heritage. | The Zoning Enabling Act gives local government the legal authority to regulate development and land use to promote public health, safety, and general welfare. The uncertainty of the guidelines for design review adopted under a zoning ordinance meet that requirement and could be challenged in court. |

In 2019, the Office of National Resources and Environmental Policy and Planning announced multiple historic districts in the country. The announcement introduced policies, area boundaries, tax reductions, and the new role of a historic commissioner, who, in conjunction with an urban planner, would determine the preservation plan criteria. Historic district establishment creates better opportunities for FBCs' implementation in these historic districts than in other areas that rely only on town planning and an urban planner's broad guidelines [23,24].

For this research, part of Phuket's Old Town was selected as a pioneer area for the FBCs' implementation. The area, which is currently in the process of preservation plan policy writing, is a historic district that retains distinctive Sino-Portuguese architecture. If these traits were to be altered or diminished, the effects on the public perception of the area would be significant. Therefore, when architectural standards and design guidelines are proposed for the district, it will be easier for stakeholders to settle on an appropriate form of architecture and establish which architectural elements are to be preserved and restored and which are to be demolished. There is also an indication of what would occur should the preservation not take place soon [25].

After applying FBCs in the historic district area of Phuket, new architectural guidelines will need to be designed that are congruent with the town's identity. These guidelines will be an essential tool for historic commissioners and urban planners to support the policy vision of the preservation plan.

## 2. The Study Area: Phuket's Old Town

### 2.1. The Study Area

Phuket province is located in the southern part of Thailand and is the only province in the country that is an island. Although its area is only the 76th largest of all of the provinces, Phuket generates the second-highest amount of income, accounting for 20% of Thailand's GDP (second only to Bangkok). More than 92% of the revenue generated was from the tourism industry. Therefore, tourist attraction is a critical strategy for income generation. Besides the natural attractions that are the main tourist destinations in Phuket, the Sino-Portuguese architecture makes Phuket Old Town district another popular spot that tourists visit regularly [26].

Phuket's old town, a historic preservation area of Phuket, was selected as the study area. It is located in the southern region of Phuket, at 7°53′24″ N 98°23′54″ E. The old town boundary is the lowland in the middle of Phuket, and the area covers approximately 2.75 km². Phuket's old town was officially announced as a historic preservation district by the National Resources and Environmental Policy and Planning in 2019 [25]. Phuket's old town municipality is a provincial administrative unit [11]. The boundary contains four main roads, Dibuk, Talang, Phangnga, and Rassada, which together cover an area of 2.76 km². The area has a culturally enriched background and sense of history, it is multicultural, and its architecture is influenced by a Sino–Portuguese style. Sino–Portuguese architecture has been present for more than 100 years. It is a cultural influence that Phuket received from Penang, Malaysia, in the late 19th to early 20th century.

In Phuket's Old Town district, historic buildings' determination criteria are age, architectural form, aesthetic rarity, and vernacular properties, as written in the Thailand Fine Art Department's guidelines. In total, there are 388 historic buildings—60.50% of all buildings in Phuket's Old Town district, as shown in the map on the right in Figure 1. Data from an on-site building survey on building transformation indicate 64.80% façade alteration, 19.40% natural degradation, 11.80% changes caused by visual obstruction, and 4.00% inappropriate building usage. This shows that the most critical factor affecting the transformation of Phuket's Old Town building is façade alteration. This is most likely due to the general urban policy that suggests only broad area development guidelines such as building height and building setback, without any specific guidance on architectural standards. Should no explicit design guidelines for the area be introduced soon, the Sino-Portuguese architecture would surely vanish [27].

Around 70% of all building styles in the area are shophouses. The shophouse was initially designed as a place for a family for both living and doing business. The inhabitants use the front part of the building for trading and the rest, including the upper floor, as a private home. A shophouse can be identified by its six characteristic styles: Localization Style Step 1; Localization Style Step 2; Eclectic Style; Localization Style Step 3; Art Deco Style; and Early Modernism Style (see the characteristics of shophouses, shown in Figure 2) [23,24,26]. For more details on building use, building height, and the characteristic styles and distribution of the shophouse, see Figures 3 and 4.

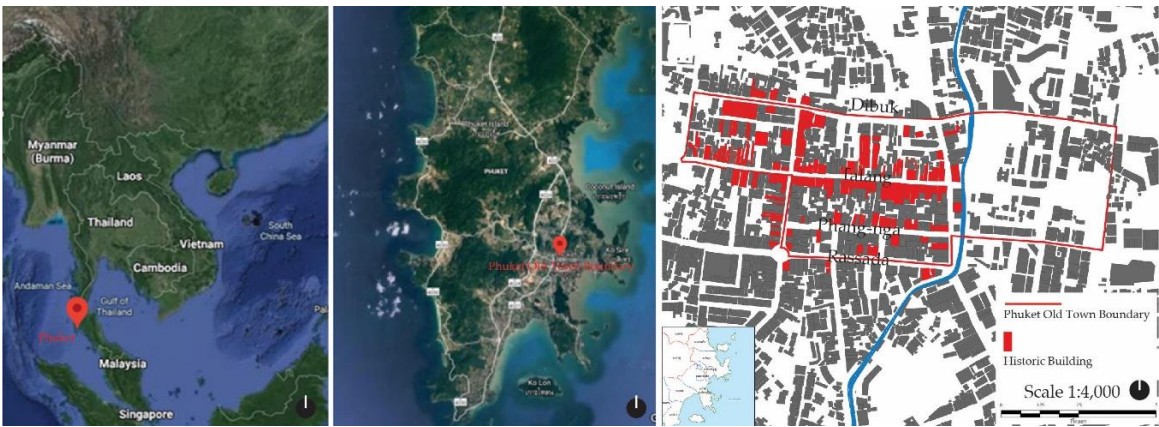

**Figure 1.** Map of Phuket's old town, Thailand.

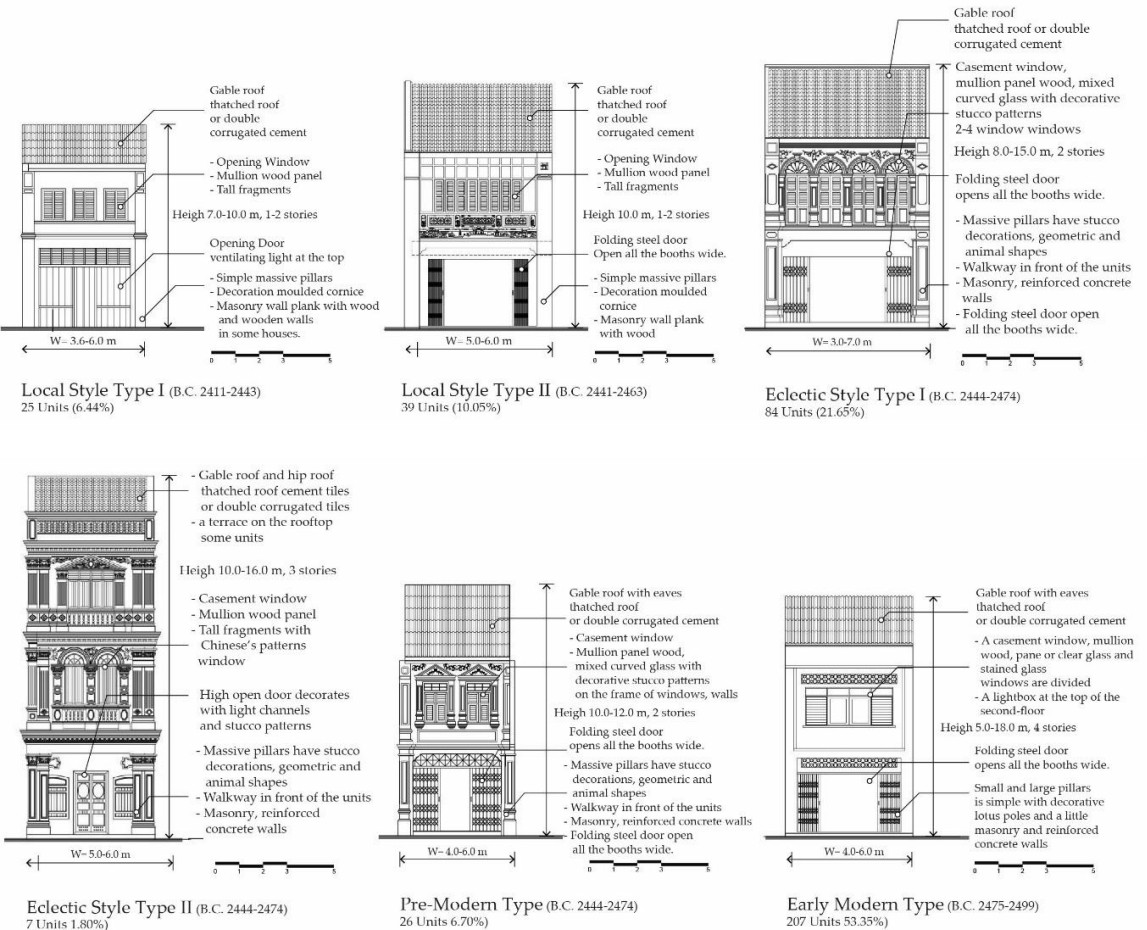

**Figure 2.** The characteristics of various types of Sino-Portuguese style in Phuket Old Town, Thailand.

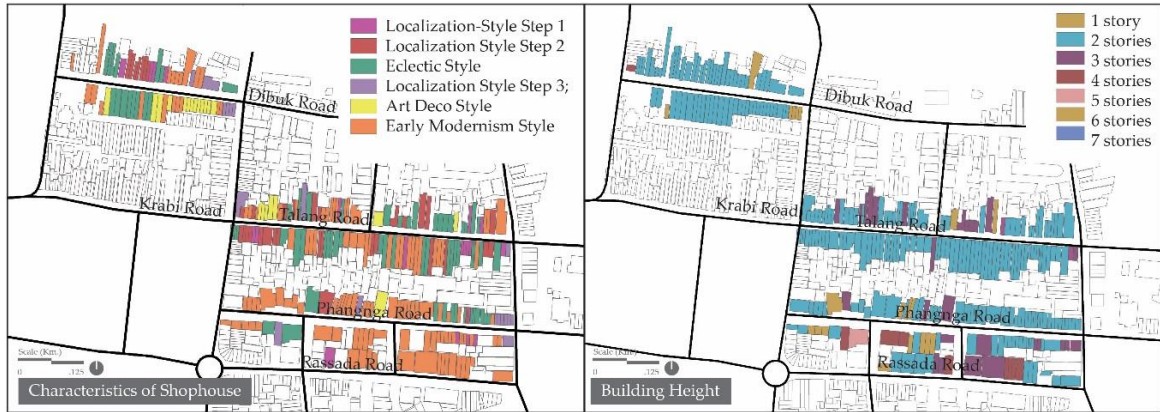

**Figure 3.** The characteristic styles and distribution of Sino-Portuguese buildings (left) and the distribution of building height (right) in Phuket's old town, Thailand.

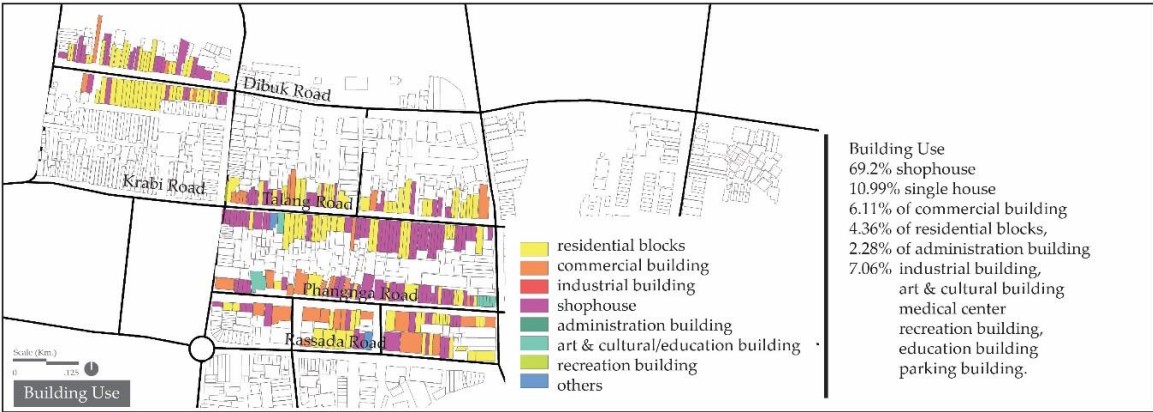

**Figure 4.** Location map of building uses in Phuket's old town, Thailand.

## 2.2. Existing Codes and Regulation Analysis

The goal of our research was to evaluate how FBCs can help to address the existing urban planning problems in Thailand to encourage the revision of codes and regulations. The study area is controlled by the Comprehensive Plan [11], the National Resources and Environmental Policy and Planning codes [25], and the Building Code [12]. These focus only on building use and a number of parameters such as setback, building height, and lot width. The old town had a close association with heritage and cultural context. Today, however, it is undergoing a tremendous transformation in its physical form and character with the new infill renovation, resulting in fewer building details and a lack of design guidelines for the area [16,19].

More than 90% of Thailand's existing laws were written as complex descriptive statements and phrases. Sometimes, experts are needed to decipher their meaning, which could pose an obstacle to local bureaus that lack sufficient knowledge or to citizens who might possess little to no knowledge of the law and regulations. On the other hand, the regulations in FBCs are written in a comprehensible and succinct fashion. FBCs explain the details of regulations in a visual manner suitable for all audiences; no prior urban planning or law knowledge is required. An example of previous laws is given on the left and an illustration of the plan from FBCs is shown on the right in Figure 5.

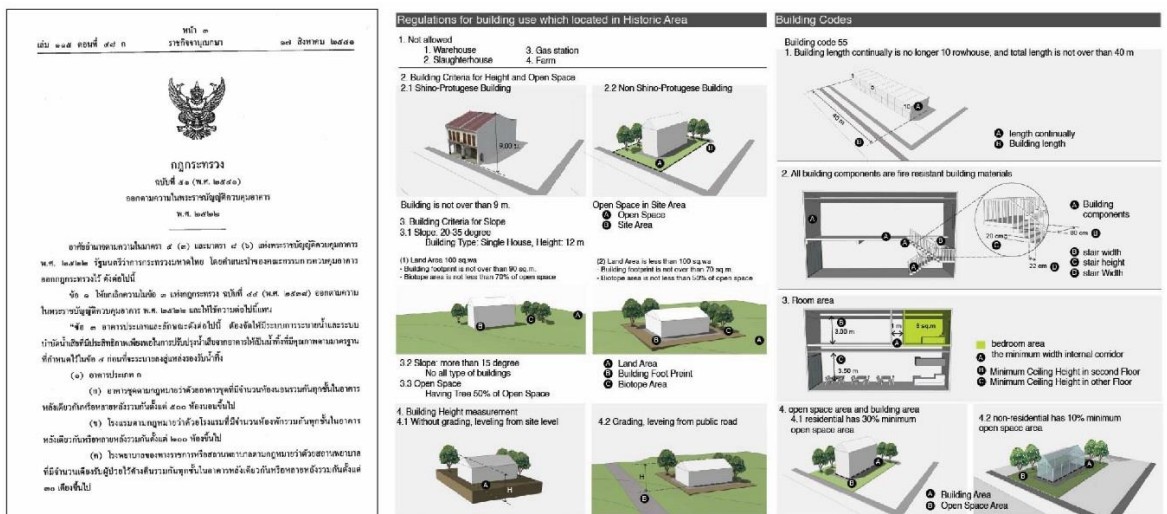

**Figure 5.** Illustration of existing codes and regulations applied to the study area.

## 3. Research Methodology

### 3.1. Data Collection

The data collection was divided into three parts, from the following FBCs components [1,2]:

(1) Documenting, Place—Communication Assessments: a combination of the micro and macro scales, which consists of Sino-Portuguese style characteristics, building placement, frontage types, building materials and elements, and building use [11,12,25].

(2) Vision, People—Communication Vision: in-depth interviews to ascertain a community vision from the three stakeholders, i.e., government, private companies, and people who own buildings in the study area. This provided a clear understanding of the intentions for the future of the community at the end of the study.

(3) Visualization, Prosperity, Vision of Growth, Assembling and Formatting, Progress—Plan Implementation: to transfer all processes into the final Form-Based Code document.

### 3.2. Data Analysis

In this research, we aimed to understand the relationship between various historical contexts and FBCs. The result shows that FBCs have the potential to enhance a historic area's character and vibrancy by improving the façade design. This research selected sample groups of buildings on Dibuk and Talang streets, which are considered main roads; a total of 240 units out of the 388 historic buildings in Phuket's Old Town were assessed. Comparisons of the building façades' physical appearance between two case studies will be carried out under the following conditions:

(1) Baseline Case—A simulation of existing building façades that is the result of four existing laws: Building Use, Setbacks, FAR and OSR in Building Codes, National Resources, Environmental Policy and Planning Codes [11,12,25].

(2) Proposed Cases—Cases that have successfully implemented FBCs. We will be studying three factors from both cases' Building Standard, Building Use, Building Placement, and Building Materials, to write a proposal for the baseline case's building façade alteration to fit the Sino-Portuguese architecture format shown in Figure 2.

The buildings' physical simulation results from both cases will be evaluated by three groups of stakeholders—a total of 116 individuals. These groups encompass government representatives, private companies, and the residents of Phuket's Old Town. Data collection will be done through personal

interviews, twice for each focus group, before the three joint colloquiums every group is required to attend. The question design for the interview consists of the four following topics:

(1) Residents' satisfaction and the existing building suitability.
(2) The ease with which the building façade can be altered according to the laws and regulations.
(3) Public awareness of the importance of the collaboration between the government, private companies, and the residents in the establishment of a preservation plan.
(4) Restrictions and comments on the implementation of FBCs in Phuket's Old Town.

The formats of the answers will be both quantitative (gauging the suitability of the proposed case using the factors derived from previous studies) and qualitative (open-ended questions). The experts in the historic preservation field, architectural field, and Thai urban planning code will re-analyze the evaluation results to propose guidelines for developing the physical appearance of the buildings in Phuket's Old Town. The findings will increase the design guidelines' feasibility and sustainability. In the final process, these design proposals of building façades will be added to an action plan that is congruent with the vision of the preservation plan.

The research methodology and procedure are summarized as a diagram in Figure 6.

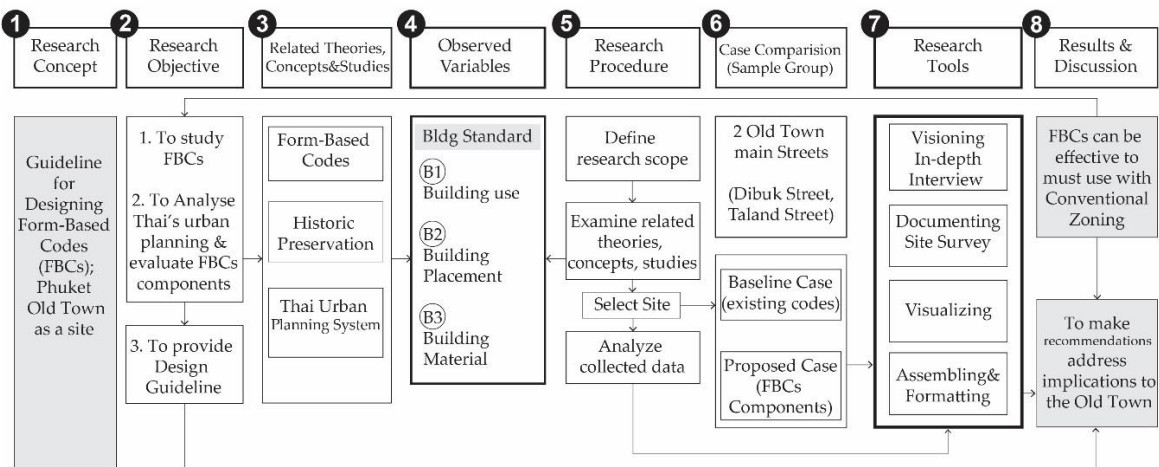

**Figure 6.** Research methodology and procedure framework.

## 4. Results and Discussion

### 4.1. Documenting

Table 2 shows the relationship of the existing codes, the building form standards, and the observed variables found in the study area. It demonstrates the relationship between existing codes and Form-Based Codes by showing its compatibility, rated for both intention and details. The laws and regulations are compatible in terms of the three intentions, especially in the case of building uses that encourage mixed-use housing and commercial buildings (shophouses). As for the building height and building material regulation determined by the Building Codes, National Resources, Environmental Policy, and Planning Codes, only generic measures, such as setbacks, are defined. There was no determination of the number of parameters and other building details, especially characteristic-defining details of the building.

**Table 2.** The relationship of existing codes, building form standards, and observed variables.

| Observed Variables | Existing Thai Codes | Form-Based Codes | Compatibility of Codes | |
|---|---|---|---|---|
| | | | Intentions | Details |
| B1—Building Use | Comprehensive plan | Allowed building use | Good | Good |
| | Allow residential, commercial, government building, art, and cultural building and recreation area | | | |
| B2—Building Placement | Building codes /National Resources: for Sino–Portuguese and regular building | Building height, building height, parallel grid lines, repetitive elements | Good | Fair |
| B3—Building Materials | Building codes: control only fire components and no details of elements and materials | Building Façade: suggest for building elements, lighting, signage | Fair | Improve |

*4.2. Visioning*

In-depth interviews were conducted with members of the three stakeholder groups, i.e., the government, private companies, and people who own buildings in the study area, via both individual interviews and focus groups. From the government sector, we interviewed eight people, a mixture of urban planners, architects, engineers from the Division of Public Works, the Subdivision of Surveys and Designing, and the Subdivision of Disaster Prevention and Environment at Phuket's town municipality [28,29]. From private companies, we interviewed 12 people who are presidents or officers from Phuket development, and 96 residents who own buildings in the study area. Based on the opinions of more than 80% of the stakeholders, we derived a general consensus from the participants. From these interviews, there should be a specific design guideline created for each area, to preserve historic buildings and district character. The results were developed into three flagship projects as follows:

(1) The design and determination of guidelines and codes for architectural preservation and restoration that include both building façades and building placement. These guidelines and codes should be presented as options to be chosen when designing and constructing both old and new buildings.

(2) Giving the proper rights to landowners that follow design guidelines. These rights include (1) tax incentives or subsiding patronage, (2) promoting the preservation and restoration of the city's historical architecture and heritage, and (3) collecting funds to preserve, conserve, and manage building preservation projects that are related to different templates dependent on the degradation of the building, using both government and private funds. The stakeholders' visions for both government and house owners should be developed, in order to increase the organization's project and the staff's potential and ability.

*4.3. Visualizing, Assembling, and Formatting*

In this study, we aimed to create building form standards to enhance the identity of Sino-Portuguese buildings. These work like FBCs, using an illustrative plan and other vision documents. For the data collection and evaluation, we utilized a rating scale and the site survey of building components of the baseline case. In-depth interviews for suggesting FBCs were completed with four participants from the government, private companies, and residents of Phuket's old town (see Section 4.2). Afterwards, the results were evaluated by 10 experts from the historic preservation field, the architectural field, and Thai urban planning, as shown in Table 3. Building placement and building frontage were assessed comparing the baseline case and the proposed case, as shown in Figures 7 and 8.

**Table 3.** Results of building components of the baseline case.

| Item | Observed Variables | Baseline Case (%) | | | |
|---|---|---|---|---|---|
| | | DB (1) | DB (2) | TL (1) | TL (2) |
| **Total buildings (units)** | | 56 | 61 | 77 | 89 |
| **Numbers in the Baseline Cases (%)** | Historic buildings | 75.00 | 85.25 | 88.31 | 87.64 |
| | Nonhistoric buildings | 25.00 | 14.75 | 11.69 | 12.36 |
| **Type of Sino-Portuguese style in Phuket Old Town (%)** | Localization Style I | 30.77 | 7.14 | 0 | 0 |
| | Localization Style II | 30.77 | 7.14 | 16.67 | 54.55 |
| | Eclectic Style I | 28.08 | 71.43 | 0 | 45.45 |
| | Eclectic Style II | 0 | 7.14 | 66.67 | 0 |
| | Pre-Modern | 0 | 0 | 0 | 0 |
| | Early Modern | 0 | 0 | 0 | 0 |
| | N/A | 15.38 | 7.14 | 16.67 | 0 |
| **Building Components** | | | | | |
| **B1—Building Use (%)** | B1a—Residential | 30.78 | 71.44 | 25.00 | 0.00 |
| | B1b—Commercial | 15.48 | 14.28 | 16.67 | 18.18 |
| | B1c—Shophouse | 53.84 | 14.28 | 58.33 | 81.82 |
| **B2—Building Placement** | B2a—Color Scheme | Good | Good | Good | Fair |
| | B2b—Parallel Gridlines | Fair | Good | Fair | Good |
| | B2c—Repetitive Elements | Improved | Good | Fair | Fair |
| **B3—Building Materials** | B3a—Design inspiration: wall, door, windows, post | Fair | Good | Fair | Fair |

Note: DB = Dibuk Street and TL = Talang Street.

In the baseline case of Dibuk (DB) street, the results revealed that the building use (B1) is 58.84% shophouse (1) and 71.44% residential (2), with a majority of two-story buildings. Most building façades (B3) are almost completely well preserved, including walls, doors, and posts, especially those on DB (2) street. Because of this, the proposed case, which had implemented the Form Based-Code, suggests renovation guidelines that only renovate ornamental materials, building height, and roof form of the 1.12–1.13 on DB (1) street and 1.14 on DB (2) street.

Similarly, Talang (TL) street is where the most complete Sino–Portuguese buildings are located, and shophouses (B1c), in terms of building use (B1), represent 58.33% and 31.82% of the total building use in TL (1) street and TL (2) street, respectively. Most buildings have two stories. TL (1) street has a building that exceeds both the size and height of its general building. However, it is exempted, as it was placed at the corner of the road, serving as the road's landmark. The building façade materials (B3) that belongs to its period are fairly preserved, though lacking certain elements, such as a skylight on the second floor and the iron elements on the doors, which were changed to glass. The proposed case suggests renovation guidelines that change only ornamental materials, building height, and the form of the 1.12–1.15 building on TL (1) street and 1.5 and 1.10 building on TL (2) street.

The outputs were combined, and we compared the baseline case and the proposed case that adopted the building components. Three observed variables (allowed building use, building placement, and building façade) were moderately similar based on the percentage of the baseline case and the proposed case for each variable. Meanwhile, the results of the building placement, including the parallel gridline (B2b) and repetitive elements (B2c), did not match well—especially the building's form and height and the completeness of the building's elements.

In conclusion, the stakeholders and experts agreed that there should be guidelines for Sino-Portuguese architectural elements for both the renovations and infills included in the building façade development proposal in this case. This guideline will be an essential tool in the historic district's

preservation. Not only will the regulations be based on architectural standards that are congruent with the area's context and laws, but comments from stakeholders will also play an important role, and the re-evaluation of experts in each field will indicate the practicality of the plan. The plan also draws heavily on community needs rather than deriving the regulations directly from the central government, without considering the importance of on-site users. This might pose a significant problem, such as has been seen with the current Thai urban plan: the adaptation of top-down control management, combined with a lack of detailed laws—especially in special districts such as Phuket's Old Town—has not been successful [16,19].

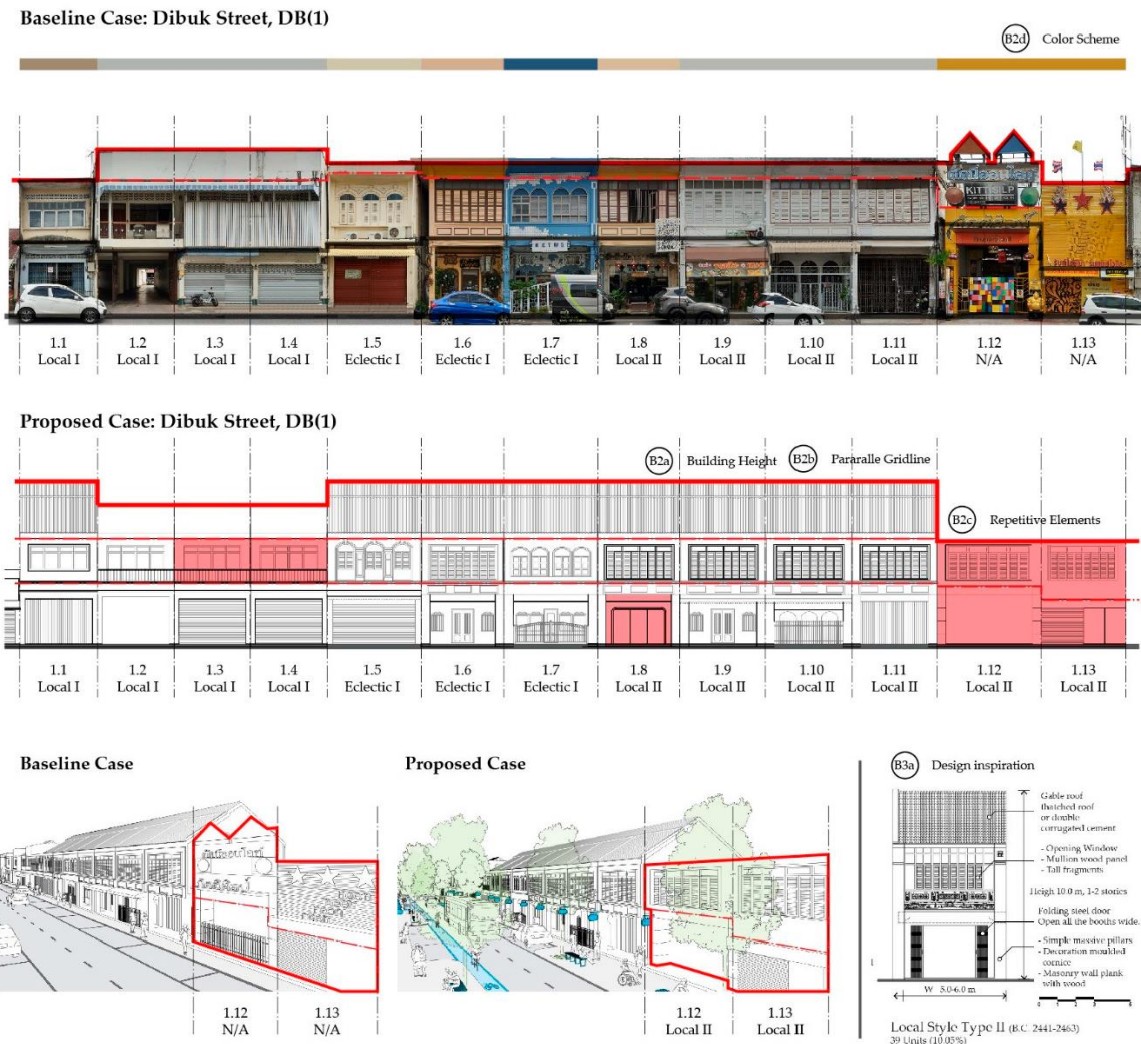

**Figure 7.** *Cont.*

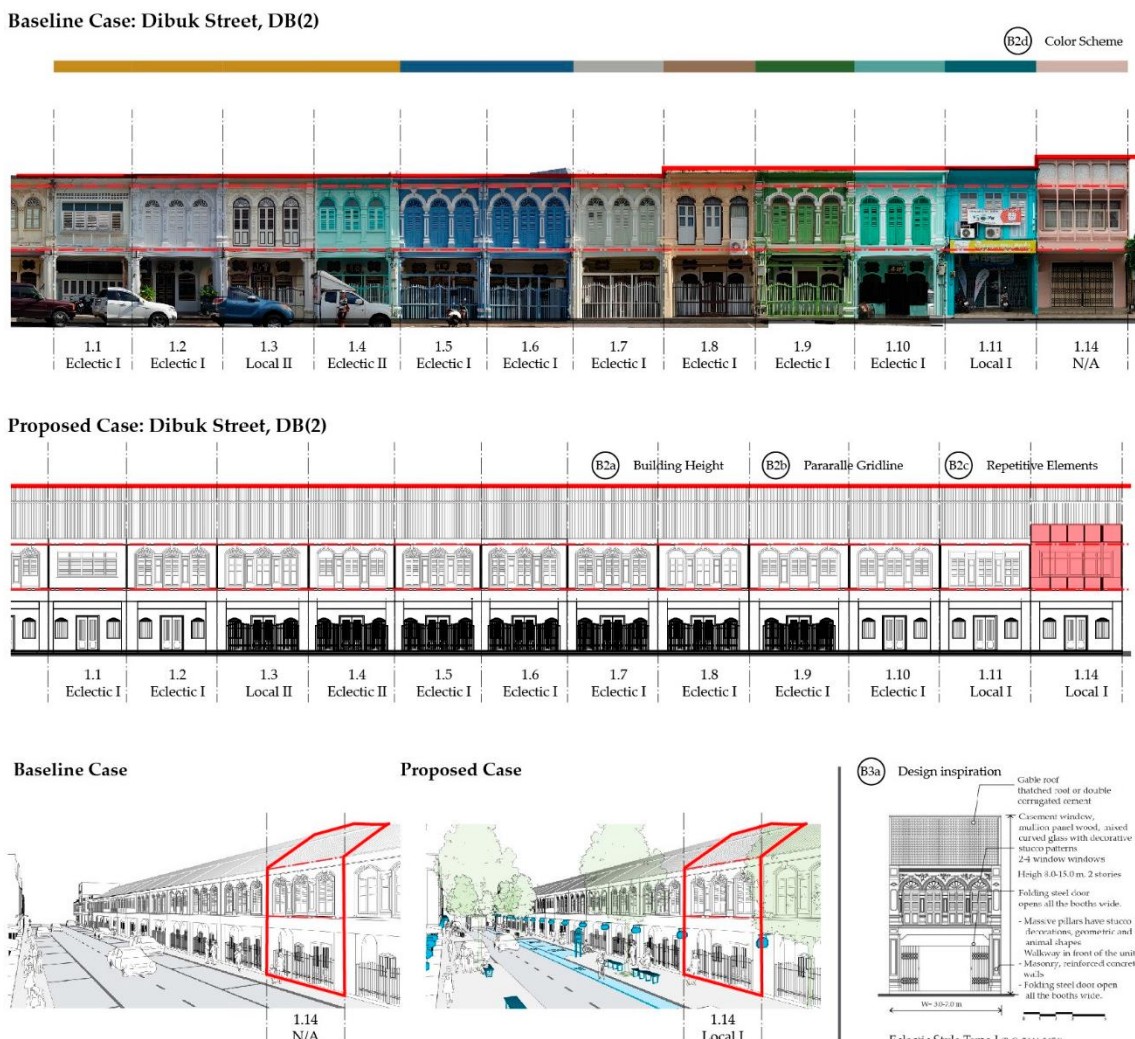

**Figure 7. Dibuk Street: Baseline Case and Proposed Case.** The illustrative façade design of the baseline case and the proposed case in Dibuk road, Phuket Old Town, Thailand.

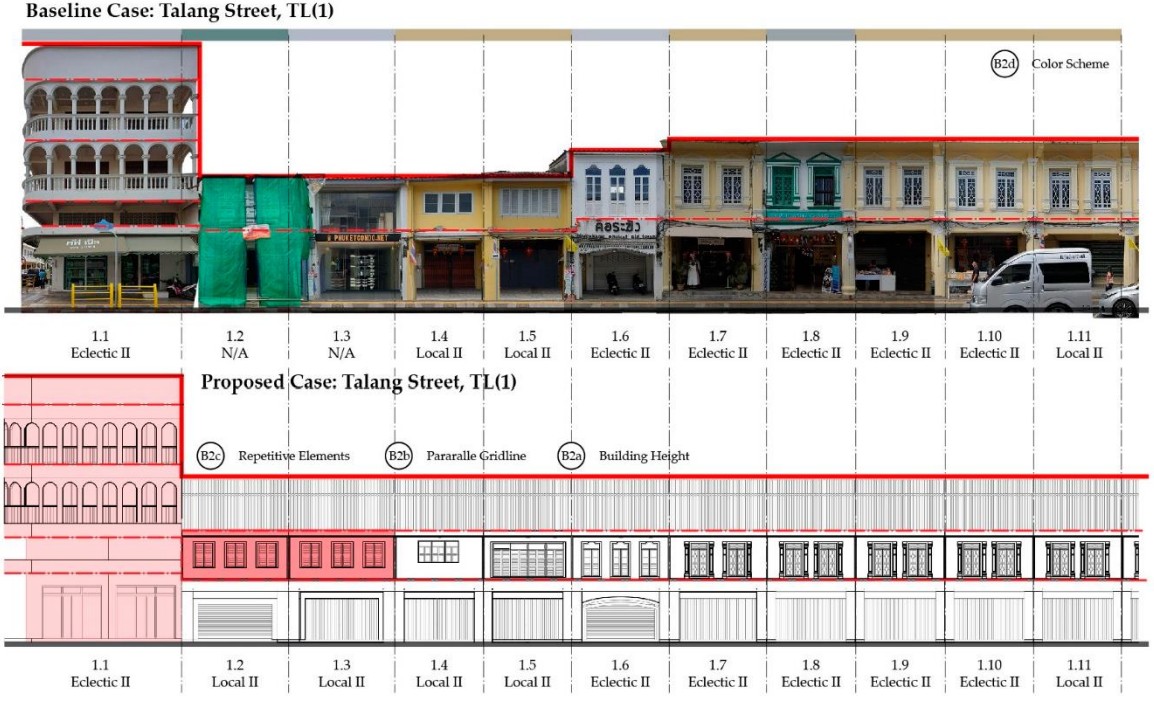

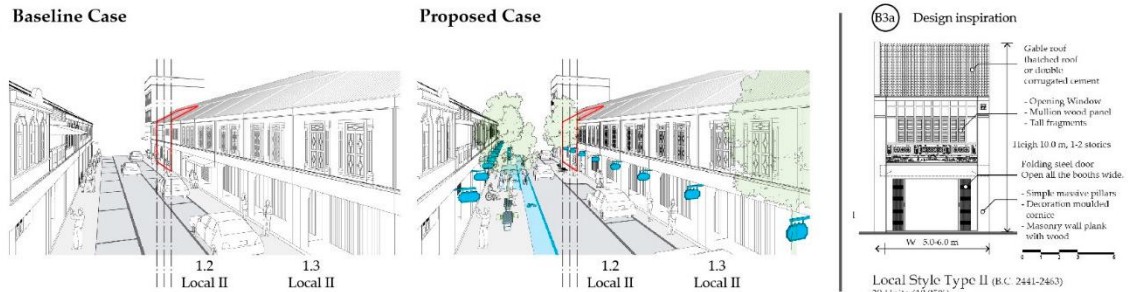

**Figure 8.** *Cont.*

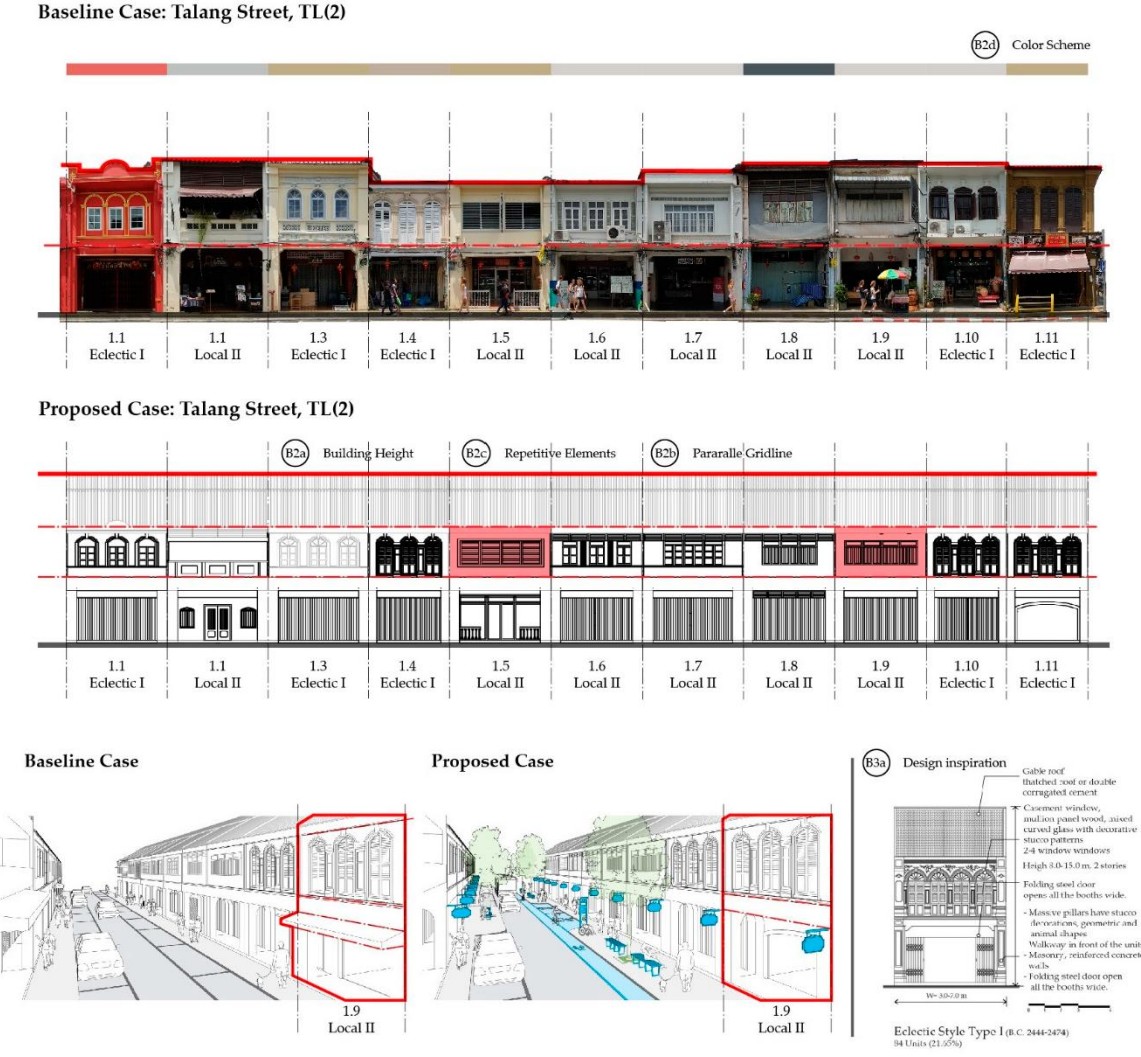

**Figure 8. Talang Street: Baseline Case and Proposed Case.** The illustrative façade design of baseline case and proposed case in Talang road, Phuket Old Town, Thailand.

## 5. Conclusions and Recommendations

### 5.1. Research Conclusion

We studied and created design guidelines for historic buildings in the study area by using building form standards and following the FBC process, that consists of documenting, visioning, visualizing, assembling, and formatting. The building form standards are primarily organized according to the period of the building's construction. The typical regulatory elements include building use, building placement, building type, and building façade. In conclusion, the majority of the buildings can successfully retain their existing shape and forms. This is so is because Phuket City's old town is a globally renowned tourist destination, and various laws and regulations were made and enforced to protect, preserve, and conserve the area. These laws and regulations include a Comprehensive Plan, a National Resources and Environmental Policy, Planning Codes, and a Building Code. Since local stakeholder relationships and networks in the area are firm, multiple meetings and activities exist to support and encourage the constant social awareness of the importance of its cultural heritage. However, since Phuket's old town and many other Thai historical districts have yet to establish design guidelines for contemporary architectural details and elements, there is a tendency for unwanted alterations—or, even worse, the disappearance of the existing building character [30,31].

To encourage public participation, Form-Based Codes work well in established communities. They adequately define and codify a neighborhood's existing characters. Historic building types can be easily replicated, promoting an infill that is compatible with the surrounding structures. FBCs allow all stakeholders to have a say in the preservation and design of the buildings, which leads to a higher comfort level as regards the greater density of building use, for instance. Form-Based Codes offer flexibility that allows new construction and infill to be tailored by multiple property owners. They can regulate development at the individual scale of a building or lot [32,33].

## 5.2. Recommendation on Thai Urban Planning in Historic Districts

Form-Based Codes are controversial due to the public's involvement and are the key to protecting historic districts and resources. Each area is specific and may not have enough awareness and concern for the impact on the surrounding area and sense of place. The degree of power and responsibility of the historic preservation commission is concerning. Phuket's old town does not have a specific plan or local historic codes. They might establish a historic district commission (HDC) and a training program to deal with sensitive historic resources. An HDC should organize and clarify the working process regarding communication assessments, community vision, and plan implementation [9,19].

Importantly, it is easier for a non-planner to determine whether compliance is achieved. The FBC documents are much shorter, more concise, and visually accessible and readable than conventional zoning documents. We suggest changes in the details of the process of documenting and envisioning to achieve and enhance community assessment and vision as shown in Figure 9.

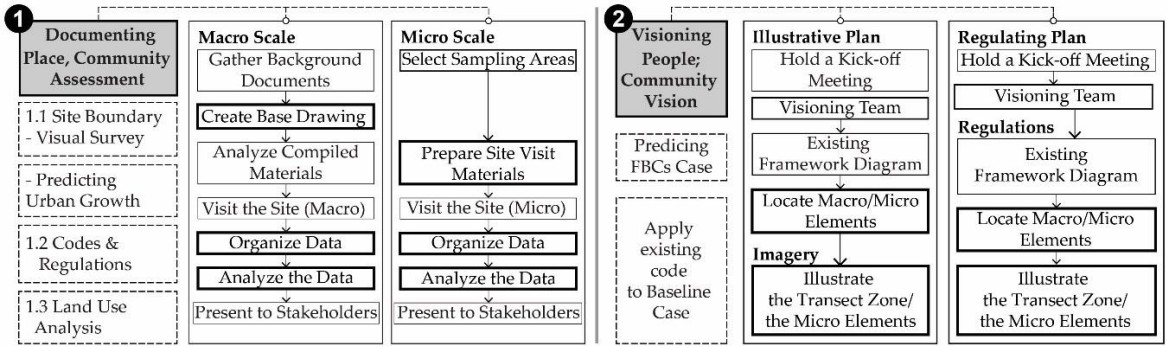

**Figure 9.** Suggestions for the documenting and visioning process of the Form-Based Code (FBC) process.

## 5.3. Recommendations

Thailand has no local historic commissioner with specialized expertise in historic preservation practices, who could support the urban community and promote public participation and input. A history commissioner is crucial in creating design standards. They have to create a preservation plan and historic preservation ordinance before adopting Form-Based Codes. A survey of the city's historic resources also needs to be conducted. It is important to take a record of all the extremely significant historic resources. The historic preservation commission should have an important role in developing the codes and offer their professional opinion on the design standards created [19].

**Author Contributions:** Conceptualization, K.M.; methodology, P.L., K.M.; project administration, P.L.; supervision, P.L.; validation, P.L.; visualization, K.M.; writing—original draft, K.M.; writing—review & editing, P.L. All authors have read and agreed to the published version of the manuscript.

**Funding:** The authors are grateful to the Royal Golden Jubilee Ph.D. (RGJ-Ph.D.) Program (Grant No. PHD/0205/2558), under the Thailand Research Fund (TRF), for funding this research.

**Conflicts of Interest:** The authors declare no conflict of interest.

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
