# Peer review of "Incorporating Form-Based Codes into the Design-Based Approach to Historic Building Conservation in Phuket, Thailand"

_sustainability, doi:10.3390/su12093859_

Round 1

Reviewer 1 Report

The introduction to the article does not provide a clear idea of research questions and the significance of the study. It is not clear what the problem with existing codes in relation to the historic area selected. The Figure 5 does not clearly explain it. The method of the research is not clear as written; for example, no information/justification is given on the sample size, sample selection, questions asked, and how data were collected are given. No clear operationalization of what it means a ‘baseline case’ and a ‘proposed case’ and how these cases are selected are given in the article. It is also not clear what was found out from baseline cases and how those findings are applied to the proposed case, and why the findings from the baseline cases are considered significant to develop a FBCs. It is not clear what FBCs are proposed for Phuket’s historic area.

Reviewer 2 Report

This research is about establishing a FBC framework restoring historic architectural motives as a distinctive style characterizing a old town of Thailand to preserve a sense of place. The motivation and targeted issue seem to be interesting but it may need the following enhancement for being considered to be published. 

1. Authors may need to elaborate on 'Sino-Portugease style' and various characteristic architectural styles illustrated in Fig. 2 in terms of their origins, formation and evolutionary paths, distinctive visual elements and other related facts to assure those styles are worthy of being preserved for maintaining  socio-cultural heritage.

2. In-depth interview being mentioned in section 4.2 (line 170) need to be described in detail in terms of its goal, process and method, participants and outcomes to assure the derived guidelines are based on collective consensus. 

3. Table 3 needs the following improvements. 

3-1. 'B2- Building height' component is explained as 'Building Placement', which does not possessing clear mapping. Even observed variables(Color Scheme, Parallel Gridline, Repetitive Elements) are not conceptually links to 'B2-Building height' categorization. Thus, revise or explain these would enhance the clarity of the manuscript. 

3-2. Numbers in the 'Baseline Cases' portion of Table 3 needs unit(%) specification to make clear what the numbers represent.

3-3. Again, Table 3's 'Baseline Case' section has 'Good', 'Fair', Impro' ratings in its middle segment. Whose judgement are these? How this rating could be justified? 

4.  Figure 7 and Figure 8 need to be explained in detail in terms of deriving and analyzing  base cases, feature extractions and proposed design guides.

5. It is not clear how and where in the manuscript the authors suggest or demonstrate FBC proposal targeted for Phuket's old town, thus, the suggested FBC need to be presented in a clearer manner.  

Author Response

RE: Revised Article

Journal: Sustainability (ISSN 2071-1050)

Manuscript ID:sustainability-774644

Type: Article

Number of Pages: 15

Title: Incorporating Form-Based Codes into the Design-Based Approach for Historic Building Conservation in Phuket, Thailand

Authors:Pusit Lertwattanaruk *, Kanokwan Masuwan

Dear Editor-in-Chief

Enclosed please find the detailed response to the reviewers’ comments and the revised manuscript for the paper titled “Incorporating Form-Based Codes into the Design-Based Approach for Historic Building Conservation in Phuket, Thailand” to be reviewed for publication in the journal Sustainability.
Yours sincerely,

Associate Professor Pusit Lertwattanaruk, Ph.D.

Department of Architecture,

Faculty of Architecture and Planning

Thammasat University

Khlong Luang, Pathumthani, 12121

THAILAND

Tel: 66-2-986-9434

English language and style

( ) Extensive editing of English language and style required 
( ) Moderate English changes required 
(x) English language and style are fine/minor spell check required 

The manuscript has undergone English language editing by MDPI. Please kindly find the attached Certificate of English Editing.

 ( ) I don't feel qualified to judge about the English language and style 

Comments and Suggestions for Authors

1. Authors may need to elaborate on 'Sino-Portuguese style' and various characteristic architectural styles illustrated in Fig. 2 in terms of their origins, formation and evolutionary paths, distinctive visual elements and other related facts to assure those styles are worthy of being preserved for maintaining socio-cultural heritage

Responses: 

The manuscript was revised as described in Section 2.1 to reflect this comment.

"In Phuket’s Old Town district, historic buildings’ determination criteria are age, architectural form, aesthetic rarity, and vernacular properties, as written in the Thailand Fine Art Department’s guidelines. In total, there are 388 historic buildings—60.50% of all buildings in Phuket’s Old Town district, as shown in the map on the right in Figure 1. Data from an on-site building survey on building transformation indicate 64.80% facade alteration, 19.40% natural degradation, 11.80% changes caused by visual obstruction, and 4.00% inappropriate building usage. This shows that the most critical factor affecting the transformation of Phuket’s Old Town building is facade alteration. This is most likely due to the general urban policy that suggests only broad area development guidelines such as building height and building setback without any specific guidance on architectural standards. Should no explicit design guidelines for the area be introduced soon, the Sino-Portuguese architecture would surely vanish. "

2. In-depth interview being mentioned in section 4.2 (line 170) need to be described in detail in terms of its goal, process and method, participants and outcomes to assure the derived guidelines are based on collective consensus. 

Responses: 

The manuscript was revised as described in Section 4.2 to reflect this comment.

In-depth interviews were conducted with members of the three stakeholder groups, i.e., the government, private companies, and people who own buildings in the study area, via both individual interviews and focus groups. From the government sector, we interviewed eight people, a mixture of urban planners, architects, an engineers from the Division of Public Works, the Subdivision of Surveys and Designing, and the Subdivision of Disaster Prevention and Environment at Phuket’s town municipality [28,29]. From private companies, we interviewed 12 people who are presidents or officers from Phuket development, and 96 presidents who own buildings in the study area. Based on the opinions of more than 80% of the stakeholders, we derived a general consensus from the participants. From these interviews, there should be a specific the design guideline created for each area to preserve historic buildings and district character.”

3. Table 3 needs the following improvements. 

3.1 'B2- Building height' component is explained as 'Building Placement', which does not possessing clear mapping. Even observed variables (Color Scheme, Parallel Gridline, Repetitive Elements) are not conceptually links to 'B2-Building height' categorization. Thus, revise or explain these would enhance the clarity of the manuscript. 

Responses: 

The manuscript was revised as described in Section 1.2.1 to reflect this comment.

“1.2.1 Building placement: To regulate the relationship between buildings and the public standards by considering the lot width, the building and ceiling height, the building depth, and the building lot size to understand equal building height, parallel grid lines, repetitive elements, and build-to line (BTL) regulations, whereas conventional zoning controls only the setback, floor area ratio (FAR), building coverage ratio (BCR), and open space ratio (OSR)"

3.2 Numbers in the 'Baseline Cases' portion of Table 3 needs unit (%) specification to make clear what the numbers represent.

Responses: 

The manuscript was revised as described in Table 3 to reflect this comment.

Table 3 Results of building components of baseline case.

Item

Observed Variables

Baseline case (%)

DB (1)

DB (2)

TL (1)

TL (2)

Total building (units)

56

61

77

89

Numbers in the Baseline Cases portion (%)

Historic building

75.00

85.25

88.31

87.64

Nonhistoric building

25.00

14.75

11.69

12.36

3.3 Again, Table 3's 'Baseline Case' section has 'Good', 'Fair', Impro' ratings in its middle segment. Whose judgement are these? How this rating could be justified? 

Responses: 

As previously mentioned in section 2.1 was also revised as described in Section 4.3 to reflect this comment.

“In this study, we aimed to create building form standards to enhance the identity of Sino-Portuguese buildings. These work like FBCs, using an illustrative plan and other vision documents. For the data collection and evaluation, we utilized a rating scale and the site survey of building components of the baseline case. In-depth interviews for suggesting FBCs were completed with four participants from the government, private companies, and residents of Phuket’s old town (see Section 4.2). Afterwards, the results were evaluated by 10 experts from the historic preservation field, the architectural field, and Thai urban planning, as shown in Table 3.”

4. Figure 7 and Figure 8 need to be explained in detail in terms of deriving and analyzing base cases, feature extractions and proposed design guides. 

Responses: 

The manuscript was revised as described in Section 3.2 to reflect this comment.

In this research, we aimed to understand the relationship between various historical contexts and FBCs. The result shows that FBCs have the potential to enhance a historic area’s character and vibrancy by improving the facade design. This research selected sample groups of buildings on Dibuk and Talang streets, which are considered main roads; a total of 240 units out of all 388 historic buildings in Phuket’s Old Town were assessed. Comparisons of the building facades’ physical appearance between two case studies will be done under the following conditions:

  1. Baseline Case—A simulation of existing building facade that is the result of four existing laws: Building use, Setbacks, FAR and OSR in Building Codes, National Resources, Environmental Policy and Planning Codes [11,12,25].
  2. Proposed Cases—Cases that have successfully implemented FBCs. We will be studying three factors from both cases’ building standard, Building Use, Building Placement, and Building Materials, to write a proposal for the baseline case’s building facade alteration to fit the Sino-Portuguese architecture format shown in Figure 2.

Building physical simulation results from both cases will be evaluated by three groups of stakeholders—a total of 116 individuals. These groups encompass government representatives, private companies, and the residents of Phuket’s Old Town. Data collection will be done through personal interviews, twice for each focus group, before the three joint colloquiums every group is required to attend. The question design for the interview consists of the four following topics:

  1. Residents’ satisfaction and the existing building suitability.
  2. The ease with which the building facade can be altered according to the laws and regulations.
  3. Public awareness of the importance of the collaboration between the government, private companies, and the residents in the establishment of a preservation plan. 
  4. Restrictions and comments on the implementation of FBCs in Phuket’s Old Town

Formats of the answers will be both quantitative (gauging the suitability of the proposed case using the factors derived from previous studies) and qualitative (open-ended questions). The experts in the historic preservation field, architectural field, and Thai urban planning code will re-analyze the evaluation results to propose guidelines for developing the physical appearance of the buildings in Phuket’s Old Town. The findings will increase the design guidelines’ feasibility and sustainability. In the final process, these design proposals of the building facade will be added to an action plan that is congruent with the vision of the preservation plan.

The manuscript was revised as described in Section 4.3 to reflect this comment.

In conclusion, the stakeholders and experts agreed that there should be guidelines for Sino-Portuguese architectural elements for both the renovations and infills included in the building facade development proposal in this proposed case. This guideline will be an essential tool in the historic district’s preservation. Not only will the regulations be based on architectural standards that are congruent with the area’s context and laws, but comments from stakeholders will also play an important role, and the re-evaluation of experts in each field will indicate the practicality of the plan. The plan also draws heavily on community needs rather than deriving the regulations directly from the central government without considering the importance of on-site users. This might pose a significant problem, such as has been seen with the current Thai urban plan: the adaptation of Top-Down Control management, combined with a lack of detailed laws, especially in special districts such as Phuket’s Old Town, has not been successful”

5. It is not clear how and where in the manuscript the authors suggest or demonstrate FBC proposal targeted for Phuket's old town, thus, the suggested FBC need to be presented in a clearer manner.

Responses: 

The manuscript was revised as described in Section 1.4 to reflect this comment.

“In 2019, the Office of National Resources and Environmental Policy and Planning announced multiple historic districts in the country. The announcement introduced policies, area boundaries, tax reductions, and the new role of a historic commissioner, who, in conjunction with an urban planner, would determine the preservation plan criteria. Historic district establishment creates better opportunities for FBCs’ implementation in these historic districts than in other areas that rely only on town planning and an urban planner’s broad guidelines [23,24].

For this research, part of Phuket’s Old Town was selected as a pioneer area for the FBCs’ implementation. The area, which is currently in the process of preservation plan policy writing, is a historic district that retains distinctive Sino-Portuguese architecture. If these traits were to be altered or diminished, the effects on the public perception of the area would be significant. Therefore, when architectural standards and design guidelines are proposed for the district, it will be easier for stakeholders to settle on an appropriate form of architecture and establish which architectural elements are to be preserved and restored and which are to be demolished. There is also an indication of what would occur should the preservation not take place soon [25].

After applying FBCs in the historic district area of Phuket, new architectural guidelines will need to be designed that are congruent with the town’s identity. These guidelines will be an essential tool for historic commissioners and urban planners to support the policy vision of the preservation plan

The manuscript was also revised as described in Section 2.1 to reflect this comment.

(the first paragraph) Phuket province is located in the southern part of Thailand and is the only province in the country that is an island. Although its area is only the 76th largest of all of the provinces, Phuket generates the second-highest amount of income, equating to 20% of Thailand’s GDP (second only to Bangkok). More than 92% of the revenue generated was from the tourism industry. Therefore, tourist attraction is a critical strategy for income generation. Besides the natural attractions that are the main tourist destinations in Phuket, the Sino-Portuguese architecture makes Phuket Old Town district another popular spot that tourists visit regularly.

(the third paragraph) In Phuket’s Old Town district, historic buildings’ determination criteria are age, architectural form, aesthetic rarity, and vernacular properties, as written in the Thailand Fine Art Department’s guidelines. In total, there are 388 historic buildings—60.50% of all buildings in Phuket’s Old Town district, as shown in the map on the right in Figure 1. Data from an on-site building survey on building transformation indicate 64.80% facade alteration, 19.40% natural degradation, 11.80% changes caused by visual obstruction, and 4.00% inappropriate building usage. This shows that the most critical factor affecting the transformation of Phuket’s Old Town building is facade alteration. This is most likely due to the general urban policy that suggests only broad area development guidelines such as building height and building setback without any specific guidance on architectural standards. Should no explicit design guidelines for the area be introduced soon, the Sino-Portuguese architecture would surely vanish. 

---------------------------------------------------------------------------------------

Certificate of English Editing

Reviewer 3 Report

This topic is important for the protection of cultural heritage, but the manuscript needs to be improved. The basic recommendations apply to redrafting the text. The manuscript is too synthetic - it is more a technical paper than a full-length article.

It is necessary to expand the introduction. There are few references to literature in the introduction. The authors have not sufficiently described how Form-Based Codes are used in other countries, what results it brings (especially in Asian countries).

In point 1.3. "Problems in Thai Urban Planning System" authors should describe the urban planning (spatial planning) system in Thailand. This will help the reader understand the need for new urban planning tools.

The research methodology consists of three stages: Documenting - Vision - Visualization. The vision is based on interviews. The authors do not write much about interviews - it is not known what government institutions and private companies have participated in, how many owners, their scale, their form, etc. In section 4.2. "Visioning" authors write only about "80% of the stakeholders" - this is all  information. The methodology must accurately describe this.

Thepurpose described in the summary and research methodology is different.

In point 4.1. "Documenting" - existing the laws and regulations should be described in more detail - in this paragraph or in the introduction.

The literature is poor - it needs to be improved.

References have double numbers - references should be formatted as in the editorial guidelines.

The drawings (maps) should show north direction and scale. Figures 2 and 7 should also have a scale.

Round 2

Reviewer 1 Report

No specific comment.

Reviewer 3 Report

It is much better now - the manuscript has more scientific value.